# Australian Mental Health Consumers’ Experiences of Service Engagement and Disengagement: A Descriptive Study

**DOI:** 10.3390/ijerph181910464

**Published:** 2021-10-05

**Authors:** Sharon Lawn, Christine Kaine, Jeremy Stevenson, Janne McMahon

**Affiliations:** 1Lived Experience Australia Ltd., Adelaide, SA 5070, Australia; ckaine@livedexperienceaustralia.com.au (C.K.); jmcmahon@livedexperienceaustralia.com.au (J.M.); 2College of Medicine and Public Health, Flinders University, Adelaide, SA 5042, Australia; Jeremy.stevenson@flinders.edu.au

**Keywords:** mental health, primary care, mental health services, consumers, engagement, disengagement, re-engagement

## Abstract

Mental health issues are a severe global concern with significant personal, social, and economic consequences and costs. This paper reports results of an online survey disseminated across the Australian community investigating why people with mental health issues choose particular mental health services over others, what causes them to disengage from services, and what factors and qualities of services are important to consumers to support their continued engagement or re-engagement with mental health services. The importance of GPs was evident, given their key role in providing mental healthcare, especially to those referred to as “the missing middle”—consumers with mental health issues who fall through the gaps in care in other parts of the healthcare system. The study found that many respondents chose to engage with mental healthcare providers primarily due to accessibility and affordability, but also because of the relational qualities that they displayed as part of delivering care. These qualities fostered consumers’ sense of trust, feeling listened to, and not being stigmatized as part of help seeking and having their mental health needs met. Implications for education and practice are offered.

## 1. Introduction

Mental health issues are a severe global concern with significant personal, social, and economic consequences and costs. People with mental health issues are more likely to commit suicide and experience discrimination and stigma, community disconnection, and social isolation. They are also more likely to be impacted by social determinants of health such as unemployment, homelessness, poverty, family violence, disrupted education pathways, and disproportionately high rates of physical health comorbidity, which impact their overall wellbeing, capacity for recovery, and quality of life [1].

In addition to these personal and social consequences and costs, health systems in many countries struggle to meet the demands of providing effective mental healthcare. For example, the global cost of depression and anxiety is estimated to be USD 1 trillion per year [2]. Additionally, the prevalence of mental health problems is increasing over time [3], notably being exacerbated by COVID-19 [4], which has disrupted and put pressure on many parts of the health systems of all countries, including primary care, mental health services, and hospital emergency departments. It is even more important, therefore, that these systems operate efficiently and effectively to minimize unnecessary fragmentation and duplication and manage their finite resources, so that people with mental health issues who need support do not “fall through the gaps” in service systems. In Australia, the Productivity Commission (the Australian Government’s independent research and advisory body) and others have adopted the term the “missing middle” to describe this mental health group, as “people who need intensive community support to recover and go on with their lives...[who] fall between inpatient hospital services, and services for people with mild to moderate mental health problems” [5,6].

In Australia, general practitioners (GPs) in primary care practices continue to provide the bulk of mental healthcare to this large missing middle population, and this is similar in other countries. For example, Fleury et al. [7] (p. 19) remind us that, “in the course of a given year, about 80% of the population in industrialized countries consults a GP, of which roughly 30–40% have significant psychological symptoms.” GPs are well recognized as often being the first point of help seeking by individuals and families, assessment of mental health needs, and ongoing management and support, particularly for people with multimorbid physical and mental health conditions such as depression and anxiety [8]. However, concerns about the quality of care and capacity of primary care to meet the needs of this population are longstanding. In particular, GPs experience significant challenges (time-limited consultations, lack of specialist skills, and inadequate training) in performing suicide risk assessment and management of people who present to them with suicidal ideation [9]. Collaborative care between GPs and more specialist mental healthcare clinical and psychosocial care providers is widely recognized as the preferred model for the integration and sharing of resources and expertise and providing quality care, particularly for people with more complex care needs [10,11].

Approximately one third of people with mental health issues will seek help [2], and when they do, they are often faced with the decision of which mental health service to choose and how to access support. Numerous health professionals and types of support could be accessed, including GPs, psychologists, psychiatrists, social workers, peer support groups, and online or digital resources. Many factors are associated with help seeking for mental health concerns, including cost and availability, cultural beliefs, values and attitudes about mental illness, stigma, and past experiences of services. For some, however, this may not involve choice, due to legally imposed treatment or limited availability or affordability of support options. Central to these concerns about engagement in mental healthcare is gaining a better understanding of the experiences of individuals seeking help, their experiences of care once they access and are receiving services, and why some disengage from services. This is particularly important given the body of evidence suggesting that those who disengage may be among those most in need of mental health support, most likely to present to emergency departments in crisis, and have worse mental health and psychosocial outcomes [12].

There has been some international research about reasons for disengagement from mental health services [12,13,14,15,16]. For example, an international review of 14 studies that looked at rates of disengagement and characteristics of people who disengage from mental health services [12] found significant variability in reported rates (4–46%). Presenting a somewhat service-centric view, this review also concluded that young age, ethnicity and deprivation, lack of insight, substance misuse, and forensic history were associated with higher levels of disengagement. However, the authors conceded that these associations were complex and multifaceted and confounded by varying definitions and poor conceptualization of disengagement, which was predominantly defined as being “drop-out” and “out of touch” from services. They noted the importance of engagement in the early stages of illness as crucial and also that there has been limited research on the therapeutic alliance between consumers and mental health professionals in relation to disengagement. An international scoping review [13] similarly proposed that consumers who were more likely to disengage were younger in age, male gender, from ethnic minority background, and with comorbid drug abuse. They also noted consumers’ desire to solve their own problems, dissatisfaction with treatment, inconvenience and practical constraints, and desire for greater control and less coercive interactions with health professionals, perceiving that they no longer needed support or that it was unhelpful.

In a UK study involving interviews with 40 adults living in the community [14] consumers reported the desire to be independent, poor therapeutic relationship, and loss of control as their main reasons for disengagement with services. A US study involving interviews with 56 adults with severe mental illness living in the community and 25 providers [15] found that consumers’ main reasons for disengagement were services not being perceived as relevant to their needs, inability to trust providers, and the belief that they were not ill. A Netherlands study with a sample of 529 adults of three community mental health teams [16] found that consumers are not as unwilling to receive services as previously thought and that the number of weeks required to have a first conversation with the support provider made the largest contribution to effective engagement; however, they stressed that 80.8% of the variation in engagement scores remained unexplained.

There is a dearth of Australian research on disengagement from mental health services. This study is, therefore, particularly timely given the growing concerns that Australia’s mental health system is “broken” and failing to meet the community’s needs; increasing calls for significant reform that ensure consumers’ perspectives are at the center of care [6,17]. It is clear that there needs to be more autonomy and involvement of mental health consumers in their own care as well as better relationships between consumers and service providers [18]. It remains unclear, however, why consumers choose particular mental health services over others, what causes them to disengage from services, and what factors and qualities of services are important to consumers to support their continued engagement or re-engagement with Australian mental health services. The aim of our study was, therefore, to help understand these processes by asking the following:(1)Why do consumers access particular mental health services over others?(2)Why do consumers disengage from mental health services?(3)What are the most common services with which consumers re-engaged?

We wanted to examine these questions to help inform where mental health resources should be directed, and how training and professional development within the primary care and larger mental health support sector can be enhanced to maximize engagement and prevent unwanted disengagement. We also felt it important to examine this topic from a lived experience research perspective (conducted by mental health consumers and carers) to ensure that people talked openly about their experiences and that we asked questions that they felt were most relevant to them. Staniszewska et al. [19], for example, in their review of 72 studies of mental health inpatient experiences found that it was unclear if consumers had been involved as researchers, either directly or in co-design of the studies, and therefore, whether the person’s voice was adequately captured. Chadwick et al.’s [20] review of studies from the US, UK, and Australia on barriers to comorbidity care also stressed the need for more research from the consumer perspective. Likewise, Smith et al. [15] argued that few studies have researched the subjective experience of disengagement from the consumers’ perspective. The question of why consumers disengage is also underexplored from a lived experience research perspective in the Australian literature. Answering these questions could shed light on how general practice and mental health services, more broadly, could be optimized to meet the needs of people with mental health issues who represent the missing middle.

## 2. Materials and Methods

We conducted secondary analyses of data from an online survey conducted by Lived Experience Australia (LEA) examining the mental health service engagement and disengagement (the first Australian study on this topic conducted by a consumer carer advocacy organization). LEA is a national peak representative organization for mental health consumers and family carers. It was formed in 2002 with over 2000 individual consumer and carer members and friends across Australia [21].

The consumer survey questions were developed by LEA through a series of dedicated focus group discussions with mental health consumers and carers with lengthy experience and expertise in systems advocacy, drawn from across Australia. The final draft survey was then piloted by this group to finalize any concerns re format, layout, wording, ease of understanding, and so forth. These focus group discussions and pilot testing were conducted to establish face and content validity; however, time pressures to release the survey prohibited further reliability testing of the survey instrument.

Of the 42 survey questions (see Appendix A):Eight were preliminary questions seeking demographic information (e.g., age, gender, location);The main section of the survey involved 18 questions, which asked the type of services accessed in the last five years, then sought Likert-rated or yes/no responses to questions asking about services used and why, access to services, perceived quality of services and health professionals, disengagement and why, and re-engagement and with whom. Each of these questions provided the opportunity for respondents to make further qualitative comments;The main section also involved 5 qualitative questions about what would help people to stay engaged or re-engage with services, perceptions about what happens to people after disengagement, and preferences for services that are currently inaccessible;Three questions asked about the use of digital mental health services;Eight questions were specifically for respondents who held private health insurance cover.

The SurveyMonkey link was distributed to consumers and families/informal carers across Australia to LEA members (1113 with current email addresses) and by request for further electronic newsletter distribution through LEA’s consumer and carer organizational networks with other mental health state and national advocacy organizations. The survey took 30–60 min to complete, dependent on people’s willingness to provide further comments; many provided extensive comments on their experiences. In total, and despite being open for only 3 weeks (12 October 2020–2 November 2020), 535 individuals commenced the survey (404 identified as consumers and 131 identified as family carers) with approximately 60% completing all questions. Quantitative analysis used descriptive techniques.

A significant proportion of the survey also focused on collecting qualitative data to better understand consumers’ and carers’ experiences, which are captured in an accompanying report [22]. In this brief report, we only report on the consumer quantitative data and provide brief qualitative comments to help contextualize the quantitative results. Family/carer experiences and thematic analysis of consumer and carer qualitative data will be reported in further publications.

## 3. Results

### 3.1. Demographics and Survey Completion Rates

Consumer participants provided demographic details including geographic location, gender, age, cultural background, and languages spoken at home. Many consumers were located in a capital city (62.1%, *n* = 251), 30.4% in a regional center (*n* = 123) and 5.0% in rural–remote locations (*n* = 20). A majority of consumers completing the survey were female 78.5% (*n* = 317) and over 95% (*n* = 381) were aged between 20–69 years. Almost all participants, 98.3% (*n* = 397) spoke English as their main language at home; 16.5% (*n* = 66) were not born in Australia; 3.0% (*n* = 12) identified as being Aboriginal or Torres Strait Islanders.

Whilst 404 consumers commenced the survey, with a mean average completion rate of 99% for the eight demographic questions, fewer (*n* = 285) commenced questions in the main section of the survey and went on to answer the 23 questions in that section, with a mean average completion rate of 84.3% (range 63.2–100%). In order to test for differences between completers vs. dropouts, we ran a correlation between completion status and the demographic variables. There was only one significant difference found in that completers were more likely to speak English at home (r = 0.122, *p* = 0.014). However, given the very small number of consumer participants who said they did not speak English at home (*n* = 7), little if anything can be inferred from this difference.

### 3.2. Most Common Mental Health Services Accessed during the Past 5 Years

Of the mental health services most commonly accessed in the past 5 years, 285 consumers responded, and most identified GPs (69.12%, *n* = 197) or a psychologist/counsellor/therapist (65.91%, *n* = 185). Of note, 31.23% (*n* = 89) accessed peer support. Table 1 provides further detail. Respondents could select more than one service type, and the total number of options selected (*n* = 848) shows that many had accessed more than one type of service.

### 3.3. Reasons for Accessing Mental Health Services

When asked the main reason why they used those particular mental health services, the major contributing reasons given by the 281 who responded related to feeling listened to, included, safety, trust, and control (Figure 1, see Appendix A). The least important reasons related to inclusion of a family/carer, the gender of the mental health professional, and the perception that the mental health professional had clear plans/goals.

Examining further comments, 27 of 69 respondents who said they mainly engage with public mental health services overwhelmingly stated that they had no choice. Other reasons they gave included: the “devil you know”; they cannot afford private support alternatives; they were told to go there for help. Of note, no positive comments were made in relation to reasons for engagement with public mental health services.

### 3.4. Reasons for Disengagement

When asked about the main logistical reasons why they disengaged from the mental health service, many consumers indicated not receiving the right type of support, not having their needs met, and not being able to afford the service (Figure 2, see Appendix A).

Regarding the main interpersonal reasons for disengagement, many consumers indicated that they did not feel listened to, they felt judged/stigmatized, and they did not feel the health professionals included or collaborated with them (Figure 3, see Appendix A).

### 3.5. Most Common Mental Health Services Re-Engaged

For consumers who decided to disengage, the most common mental health services or health professionals with which they said they would re-engage were GPs (32.7%, *n* = 132), psychologists (30.7%, *n* = 130), and psychiatrists (21.8%, *n* = 88). Whilst respondents were not asked to explain the reason for this choice, they were asked to provide suggestions for what would help them stay engaged with health professionals or services or return to them to receive support. Almost half of consumer respondents (46%, *n* = 199) provided suggestions. Prominent themes from their qualitative comments were: quality of the relationship with providers; better trained staff; having peer workers available; consistent and coordinated support; accessibility and availability of the service; more persistent follow-up from the service; being listened to. They also commented on affordability of the services, collaboration and communication between health professionals, and being involved in decisions about their care. In their suggestions for how services could better re-engage with them, respondents emphasized the importance of directly contacting people (phone, text, email, letter, or visit) to follow up and find out why they disengaged.

### 3.6. Comparing Mental Health Providers

In addition to the a priori analyses, we intended to undertake a comparative analysis to determine if consumers preferred particular mental healthcare providers for specific reasons. However, due to a survey programming error, participants were able to select multiple options, which meant any comparisons are flawed. We note, however, that the group that selected GPs selected significantly fewer public mental health services (*n* = 78) compared to the group that selected public mental health services (*n* = 102), suggesting a trend of differences.

For participants who selected GPs as their primary supports (*n* = 192; respondents to each item ranging from 186–192), the major contributing reasons for engaging with GPs were: they listen to me (52.88%, *n* = 101); they collaborate with me (52.36%, *n* = 100); perceived say and control in making decisions (51.05%, *n* = 97); feel safe (50.00%, *n* = 95); trust them (48.17%, *n* = 92); not feeling judged or stigmatized (46.60%, *n* = 89). For participants whose primary supports were public mental health services (*n* = 98; respondents to each item ranging from 96–98), the major contributing reasons for engaging with these services were: affordability (57.29%, *n* = 55); they listen to me (40.82%, *n* = 40); limited other options (39.18%, *n* = 38); they collaborate with me (38.78%, *n* = 38); perceived say and control in making decisions (38.78%, *n* = 38); trust them (36.73%, *n* = 36) (see Appendix A).

## 4. Discussion

### 4.1. Why Consumers Choose Certain MHS

The findings of this study emphasize the importance consumers place on their interpersonal relations with mental health service providers as part of engagement. They reaffirm much of the existing international evidence, which emphasizes the importance to mental health consumers of feeling safe and respected in their interactions with health professionals and systems of care. For example, in a systematic review of 72 studies on consumer experiences of inpatient settings across 16 countries, safety, trust, respect, information provision, and family inclusion were emphasized as important [19]. An integrative review [18] (p. 171) also emphasized the importance of relationships between service users and providers as a basis for interaction and support, particularly where they focused on consumer involvement, empowerment, and decision making as fundamental recovery elements, which they concluded still needs improvement.

A qualitative study from Norway [23] examining experiences of continuity of care across community mental health services provides further insights into why consumers valued relational elements so highly. They determined five themes capturing the range of positive and negative experiences: relationship; timeliness; mutuality; choice; knowledge [23] (p. 763). They concluded that systems integration across support providers to improve communication would improve continuity of care, engagement, and mental health outcomes.

The current study affirms these themes and stresses key interpersonal reasons for engagement (trust, feeling listened to, having a say or control in making decisions, and feeling safe) and disengagement (not feeling listened to or included in care decisions and feeling judged/stigmatized). Whilst these interpersonal reasons for engagement and disengagement are clear from the consumers’ perspective, it is less clear whether health professionals have similar understandings. Findings of the current study suggest that mental health professionals should prioritize interpersonal components in their therapeutic alliance with consumers.

The high level of trust and engagement by mental health consumers with particular types of mental health professionals, as shown in the current study, may be a consequence of the Rogerian principles underpinning their practice, which has greater symmetry with consumers’ needs. For example, a systematic review of international literature on GP strategies for managing multimorbidity [24] found that GPs mitigated the adverse impacts of fragmented systems, complexity, and risk by prioritizing person-centeredness and relational continuity as part of individualizing care to the person and their preferences. These very elements were prioritized by survey participants in the current study; hence, it is unsurprising that GPs were rated so highly by consumers as their primary mental health support provider. Consumers’ desire for a strong therapeutic relationship within a partnership model of care has been noted in earlier research on engagement [14].

Martínez-Martínez et al. [25] found that health professionals’ perceived negative attitudes were a significant concern to consumers and recommended increasing shared decision making and reflexive practice to improve the therapeutic alliance with consumers. Core to achieving this as part of ongoing engagement is the development of epistemic trust. This is an individual’s willingness to consider new knowledge as trustworthy and relevant in meeting their needs and, therefore, worth incorporating into their life [26].

### 4.2. Why Consumers Disengage from MHS and How They Could Be Re-Engaged

Many respondents in the current research identified concerns related to trust and epistemic trust, in particular, in their decisions to disengage with services. They also highlighted that they disengaged because those services simply did not meet their needs, and the health professionals who they sought support from failed to develop a therapeutic alliance with them. Of note, O’Brien et al. [12] stressed the importance of therapeutic alliance, particularly, the influence of not being listened to or involved in decision making on disengagement. A US study by Smith et al. [15] comparing consumer and service provider perspectives on reasons for disengagement found clear differences in views across the two groups. Whilst consumers said they disengaged when they perceived services were not relevant to their needs, untrustworthy, or no longer needed, service providers emphasized lack of insight, stigma, and language and cultural barriers as the main reasons for consumer disengagement. That is, service providers predominantly saw the reasons for disengagement as external to them and that the reasons sat primarily with the consumer. This discrepancy in views suggests that structural stigma within some service systems is a significant barrier to engagement and a prominent reason for disengagement for many consumers.

The absence of any studies that investigate what mental health healthcare service consumers perceive as important to re-engage is noted, as is the need for further research in this specific area. Understanding this important part of the healthcare continuum is likely to offer significant insight and practical steps to overcoming siloed care and fragmented communication, which contribute to problems experienced by the missing middle. It could also assist in improving the management of scarce specialist mental health healthcare resources. In the meantime, GPs in particular are likely to continue to be the most accessible and affordable point of re-engagement for people seeking support for mental health issues as a consequence of their proximity and role within communities, their relational style, and more open and inclusive approach to care delivery.

### 4.3. Implications

Findings of this survey provide important clues as to which aspects of care would be useful to focus on in the mental health training and education for GPs and mental health professionals in other service settings, more broadly. Addressing structural barriers to care, focusing on the core importance of the therapeutic alliance and building greater symmetry between mental health consumers’ and service providers’ understandings of engagement and disengagement are needed.

Whilst GPs appear to be important mental healthcare providers of support to people with mental health issues, the Australian primary care business model is geared to short consultations and not conducive to GPs being the main coordinator of mental healthcare. One means of addressing these concerns is through the widespread introduction of lived experience peer support roles within and across services providing mental healthcare, including in primary care. Whilst mental health peer roles have been growing within specialist and non-government mental healthcare settings, and their value in influencing cultural change and reducing stigma, and improving social connections, personal recovery, and empowerment is recognized [27,28], their presence in primary care settings and in the education of mental health professionals is still limited [29].

Another means of addressing these concerns is to introduce social prescribing more widely across Australian primary care settings. Research has proposed that people with limited support networks and greater psychosocial challenges (such as those with mental health issues) may benefit the most from social prescribing, because it enables them to build support networks, develop coping mechanisms, and improve their engagement with sources of treatment [30]. However, the researchers acknowledge that effective social prescribing relies on effective communication and information sharing across organizational boundaries. To support mental health professionals to undertake social prescribing within their role, significant improvement is needed in communication and coordination of care efforts across the many different providers of mental healthcare, and how they partner with consumers [31].

### 4.4. Limitations

This research had a number of limitations. The sample of completed surveys was relatively small. Participants were mostly females, which does not reflect the more even representation of genders and mental health needs across the population, or those accessing mental health services. This suggests different methods are needed to more fully elicit the views of males. Specific ethnic, cultural, and sexual minority populations were also under-represented, and their reasons for engagement and disengagement may vary. Moore et al.’s US study of engagement among Black and Hispanic LGB young adults [32], for example, found stigma related to sexual orientation and cultural attitudes, ambivalence about treatment efficacy, lack of family support, and difficulty finding suitable and affordable care were important barriers.

Other limitations relate to the survey design: reliability testing was not undertaken; respondents could opt out of answering some questions; the short length of survey time (3 weeks); the sample recruited may not reflect the broader community of people with mental health issues; findings of this Australian survey may not be generalizable to other countries. We also did not ask a specific question early in the survey about whether respondents had disengaged from any mental health services in the past 5 years. Instead, we asked which mental health services they used in the past 5 years, which may account for why 119 consumers who entered the survey did not continue beyond the demographic question section. Respondents could select more than one service type as their main provider of mental health support, which then limited more comparative analysis. Additionally, the views of younger adults (18–24 years) could not be distinguished due to how this question was asked in the survey, and people under 18 years were excluded. These limitations are important, given that mental health conditions and attempts to seek treatment and care often begin during childhood, adolescence, and early adulthood.

## 5. Conclusions

In Australia and internationally, failures in delivery of person-centered care, as a consequence of fragmentation of communication and collaborative care across care services, are a major contributor to people falling through the gaps [33,34]. A range of initiatives have been trialed to improve information sharing and communication across systems. However, in the absence of significant reform to the Australian mental health system across primary, secondary, and tertiary care settings, people with mental health issues who sit with the “the missing middle” and their family carers will likely continue to be the primary systems navigators of mental healthcare. They will continue to seek out health professionals and services that are more accessible, affordable, and available to hear their concerns; otherwise, they may disengage and attempt to manage without these services.

## Figures and Tables

**Figure 1 ijerph-18-10464-f001:**
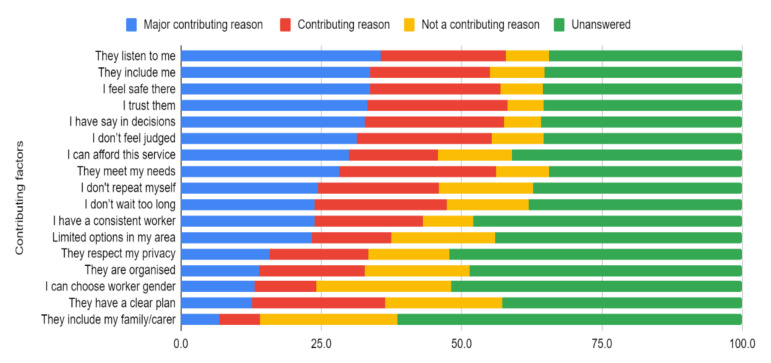
Frequency of reasons for accessing mental health services (*n* = 281 answered).

**Figure 2 ijerph-18-10464-f002:**
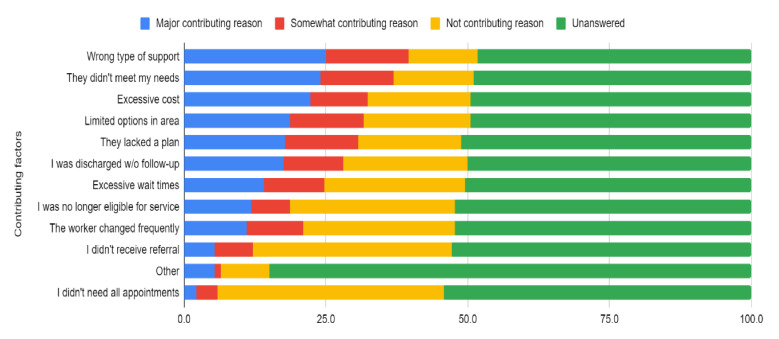
Frequency of logistical reasons for disengagement from mental health services (*n* = 223 answered).

**Figure 3 ijerph-18-10464-f003:**
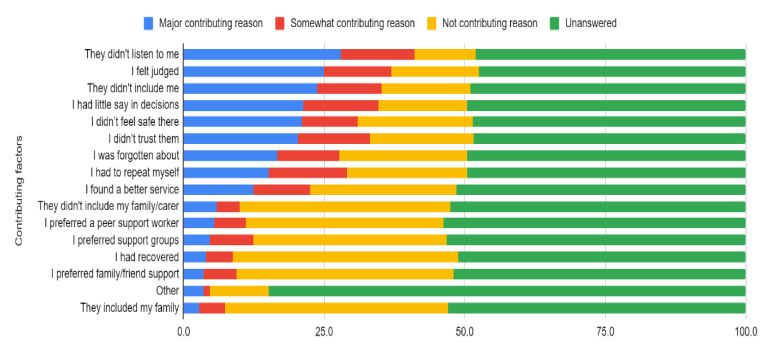
Frequency of interpersonal reasons for disengagement from mental health services (*n* = 220 answered).

**Table 1 ijerph-18-10464-t001:** Frequency of different mental health services accessed (*n* = 285).

Mental Health Service	*N*	% of Participants
My GP	197	69.12
A psychologist, counsellor/therapist	185	65.91
Public mental health services/hospitals/community teams	106	37.89
Peer support (organized or unorganized)	89	31.23
Private mental health services/hospitals	74	25.96
Online or digital resources or apps	69	24.21
Only used a private psychiatrist	57	20.00
Telehealth	54	18.95
Veteran supports	3	1.05
Other ^#^	9	3.16
Total	848	

^#^ Other included: church/spiritual coaching, assistance dog, work cover, naturopathy, employment assistance program, research trial participation, and university disability support.

## Data Availability

Access to the primary dataset on which this report is drawn is available upon request to and with the approval of the Board of Lived Experience Australia. https://www.livedexperienceaustralia.com.au/ (accessed on 4 October 2021).

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
