# Peer review of "Australian Mental Health Consumers’ Experiences of Service Engagement and Disengagement: A Descriptive Study"

_ijerph, 2021, doi:10.3390/ijerph181910464_

Round 1

Reviewer 1 Report

This article addresses an interesting topic but is essentially descriptive and based on a sample with important limitations. The discussion is interesting, but not always related to the results. Overall, the article is well written. I suggest, at the very least, that the article be reduced in length and refocused around the results. Regarding the method section, of the 535 individuals who responded to the survey, 404 were identified as consumers. The other 131 were not well identified, which makes the results related to these 131 subjects exceedingly difficult to understand. There are too many tables in the results section, adding to the length, and containing information that would be better if summarized, especially since this is essentially a descriptive study. The listing of results on p. 236 looks like a draft paper; this is not the way to present results in a scientific paper. On p. 10, the authors acknowledge that some of their results are flawed; while this is a very honest comment, it unfortunately diminishes interest in this descriptive paper and raises questions about its quality. It is also stated on p. 13 “that associations were unclear and may require further research”. This sentence is not appropriate for a scientific article, in my view. This entire paragraph should be revised for better clarity. Concerning the discussion section, it is interesting overall, but not sufficiently related to the results, and is quite long. The results and related discussion concerning individuals who responded to the survey but are not consumers is really not clear to me (nor is the second paragraph on p. 15). Subsections 4.3 and 4.4 of the discussion are interesting, but I do not see how these sections are related to the study results. The conclusion could also be improved. While the overall paper deals with an interesting subject, it does not bring much new information; the survey is limited, and the study is only descriptive. Perhaps a short summary paper would be better suited for this type of work. Minor points for revision: Some of the terminology used was not sufficiently well defined and needs to be clarified: what is “mental ill-health”, what is meant by the “missing middle”. Page 1, line 36: this sentence is also repetitive, and needs to be rewritten.

Author Response

Response to Reviewers

Reviewer 1

  1. This article addresses an interesting topic but is essentially descriptive and based on a sample with important limitations. The discussion is interesting, but not always related to the results. Overall, the article is well written. I suggest, at the very least, that the article be reduced in length and refocused around the results.

Response – We have rewritten the paper, making is much shorter, and refocused more specifically to the broad results on engagement, disengagement and re-engagement.

  1. Regarding the method section, of the 535 individuals who responded to the survey, 404 were identified as consumers. The other 131 were not well identified, which makes the results related to these 131 subjects exceedingly difficult to understand.

Response – We have now clarified this in the text ie. that the 131 were family carers.

  1. There are too many tables in the results section, adding to the length, and containing information that would be better if summarized, especially since this is essentially a descriptive study.

Response – We have removed the initial 2 figures and tables 5 and 8 altogether, replacing them with revised summary text. We have also transferred tables 2-4 to be Supplementary files (with added information about Ns, providing stacked bar charts in the manuscript, as suggested by reviewer 2.

  1. The listing of results on p. 236 looks like a draft paper; this is not the way to present results in a scientific paper.

Response – We have removed this text from the paper.

  1. On p. 10, the authors acknowledge that some of their results are flawed; while this is a very honest comment, it unfortunately diminishes interest in this descriptive paper and raises questions about its quality.

Response – we believe it is best to be as honest and transparent as possible regarding research processes, and also believe that reporting of mistakes are an important learning so that researcher who may read this work gain insights for their own future research.

  1. It is also stated on p. 13 “that associations were unclear and may require further research”. This sentence is not appropriate for a scientific article, in my view. This entire paragraph should be revised for better clarity.

Response – We have now removed the information about the further analysis (p.13) from this manuscript, given it is now a shorter report focused on the main issues of engagement, disengagement and re-engagement. We may examine that data in a future publication.

  1. Concerning the discussion section, it is interesting overall, but not sufficiently related to the results, and is quite long. The results and related discussion concerning individuals who responded to the survey but are not consumers is really not clear to me (nor is the second paragraph on p. 15).

Response – This section of the manuscript has now been removed.

  1. Subsections 4.3 and 4.4 of the discussion are interesting, but I do not see how these sections are related to the study results.

Response – Section 4.3 has now been removed. Section 4.4 has been substantially revised to shift the focus from GPs’ role.

  1. The conclusion could also be improved.

Response – We have revised the conclusion, removing the earlier focus on GPs too.

  1. While the overall paper deals with an interesting subject, it does not bring much new information; the survey is limited, and the study is only descriptive. Perhaps a short summary paper would be better suited for this type of work.

Response – We have shortened the paper significantly as suggested.

  1. Minor points for revision: Some of the terminology used was not sufficiently well defined and needs to be clarified: what is “mental ill-health”.

Response – As per reviewer 2’s suggestion, we have revised this to ‘mental health issues’ throughout the paper.

  1. What is meant by the “missing middle”.

Response – This has been defined in the introduction. Please see section at the top of p.2.

  1. Page 1, line 36: this sentence is also repetitive, and needs to be rewritten.

Response – We have revised this section as part of also shortening the paper.

Reviewer 2 Report

The authors chose an interesting topic and I greatly appreciate the community-based, participatory methods that they used to develop their measures, as that is difficult and time-consuming work. However, there are serious issues with their methods and analysis that need attention. 

Introduction:

The authors do an excellent job at describing the fragmentation of mental health services and the difficulties that prevent many people from being able to access effective care.

A small note, mental ill-health is not a common term. I suggest using mental health issues throughout for readability as this includes subclinical and clinical symptomatology.

Methods:

It would be helpful if you described a bit more about your measures instead of relying on your readers to review the Appendix. You should describe the main themes of the items, the response categories and if you conducted any analysis of their reliability or validity as part of your development of the items (e.g., factor analysis, Cronbach’s alpha). Even though you developed the scales in a highly participatory manner (which is great), you still need to establish the credibility of your measures.

Sample: How representative of the mental health population of Australia is your sample? What was your response rate?

Results:

The results need considerable attention.

Did anyone not have experience in using mental health services in the last 5 years? What was the average variety of sources?

All the tables need to include a total N as the numbers do not seem to add up to 404. It’s possible that’s due to your not including the “Unsure” category and that information should be included in your results. It is very confusing what the percentages reflect. All your tables/figures should have explanatory notes to facilitate the interpretation of your results. For example, in Table 2, 144 said that the major contributing reason was “They listen to me” but it unclear what 35.6% is a percent of, is it of the 4 response categories? The presentation of the data in tables 2, 3, 4, 6, & 7 would be better presented as figures (bar graphs). This could also facilitate other text as instead of describing in the text the main reasons for contributing or major contributing, you could have stacked bars or a total response category in a figure.

What percentage of the sample reporting disengagement from services?

The issue of not knowing what kind of provider was being rated means that you cannot make the comparisons between those who chose GPs vs public services as planned. You might be able to do those who only chose a GP, those who only chose public mental health services, and their combination but not sure how many people you would have in these categories based on your descriptions of the data.

Lastly, although you describe the demographic comparisons as exploratory, it is unclear why they need to be exploratory given your sample size. You should also be adjusting your analyses for multiple comparisons given the large number of analyses that you conduct.

Given the flaws in your data collection, this paper might be better captured as a brief report than a full manuscript.

Discussion:

The discussion raises some interesting points but presumes a level of confidence in the results that support the authors’ ideas about the value of the relationship with GPs that were not clearly made with their data (given the data collection glitch). While it was interesting to read, your data does not make a strong argument for the inclusion of section 4.3 of your discussion. You can talk about these domains more generally but the focus on GPs is not fully justified with how the results are written at present.   

Author Response

Reviewer 2

  1. The authors chose an interesting topic and I greatly appreciate the community-based, participatory methods that they used to develop their measures, as that is difficult and time-consuming work. However, there are serious issues with their methods and analysis that need attention. 

Introduction: The authors do an excellent job at describing the fragmentation of mental health services and the difficulties that prevent many people from being able to access effective care.

Response – thank you for your encouragement. We hope that our revised manuscript helps overcome these issues.

  1. A small note, mental ill-health is not a common term. I suggest using mental health issues throughout for readability as this includes subclinical and clinical symptomatology.

Response – We have revised this term as suggested throughout the manuscript.

  1. Methods: It would be helpful if you described a bit more about your measures instead of relying on your readers to review the Appendix. You should describe the main themes of the items, the response categories and if you conducted any analysis of their reliability or validity as part of your development of the items (e.g., factor analysis, Cronbach’s alpha). Even though you developed the scales in a highly participatory manner (which is great), you still need to establish the credibility of your measures.

Response – We have now included a description of the main sections of the survey and the themes of these sections. We did not undertake a factor analysis. We have now included a statement about this in the manuscript’s methods and limitations section. The questions were derived from the expertise of the dedicated consultation with established consumer and carer advocates within our mental health and community services systems across Australia, as described. The final draft survey was then piloted by this group to finalise any concerns re format, layout, wording, ease of understanding, and so forth.

  1. Sample: How representative of the mental health population of Australia is your sample?

Response – We believe the overall sample was broadly representative of the mental health population of Australia given the reach of the survey recruitment. However, we have identified a number of limitations related to gender bias towards females, and low response rates of e.g., Indigenous and non-English speaking populations.

  1. What was your response rate?

Response – We have now provided more detailed information about the response rate at the beginning of the results section.

  1. Results: The results need considerable attention. Did anyone not have experience in using mental health services in the last 5 years? What was the average variety of sources?

Response – All participants had some experience of using a mental health service of some type in the past 5 years. This could be through their contact with a GP, public mental health service, private psychiatrist, psychologist, etc. The variety of sources of mental health support were displayed in Table 1 which has now been removed and replaced with summary text.

  1. All the tables need to include a total N as the numbers do not seem to add up to 404. It’s possible that’s due to your not including the “Unsure” category and that information should be included in your results. It is very confusing what the percentages reflect. All your tables/figures should have explanatory notes to facilitate the interpretation of your results. For example, in Table 2, 144 said that the major contributing reason was “They listen to me” but it unclear what 35.6% is a percent of, is it of the 4 response categories? The presentation of the data in tables 2, 3, 4, 6, & 7 would be better presented as figures (bar graphs). This could also facilitate other text as instead of describing in the text the main reasons for contributing or major contributing, you could have stacked bars or a total response category in a figure.

Response –

  1. What percentage of the sample reporting disengagement from services?

Response – We have noted as a limitation, the absence of a specific question in the survey that would establish this. On p.5 of the manuscript (3.2), we have also noted that it is unclear whether the 119 from the original 404 who entered the survey did not answer the question about services accessed in the past 5 years because they had not accessed any mental health services during that time or because they had disengaged from services.

  1. The issue of not knowing what kind of provider was being rated means that you cannot make the comparisons between those who chose GPs vs public services as planned. You might be able to do those who only chose a GP, those who only chose public mental health services, and their combination but not sure how many people you would have in these categories based on your descriptions of the data.

Response – We made the decision not to undertake the suggested sub-analysis due to the low numbers of respondents in each of these categories. Instead, we have removed the focus on GPs from this manuscript and revised the section where we talk about making comparisons (section 3.6).

  1. Lastly, although you describe the demographic comparisons as exploratory, it is unclear why they need to be exploratory given your sample size. You should also be adjusting your analyses for multiple comparisons given the large number of analyses that you conduct.

Response – We have removed this information and section from the manuscript.

  1. Given the flaws in your data collection, this paper might be better captured as a brief report than a full manuscript.

Response – We have undertaken significant revisions and shortened the manuscript, as suggested.

  1. Discussion: The discussion raises some interesting points but presumes a level of confidence in the results that support the authors’ ideas about the value of the relationship with GPs that were not clearly made with their data (given the data collection glitch). While it was interesting to read, your data does not make a strong argument for the inclusion of section 4.3 of your discussion. You can talk about these domains more generally but the focus on GPs is not fully justified with how the results are written at present.

Response – We have removed this section from the manuscript and also shifted the focus away from GPs’ specific role.
